# Use of Selected Lactic Acid Bacteria and Carob Flour for the Production of a High-Fibre and “Clean Label” Plant-Based Yogurt-like Product

**DOI:** 10.3390/microorganisms11061607

**Published:** 2023-06-18

**Authors:** Chiara Demarinis, Marco Montemurro, Andrea Torreggiani, Erica Pontonio, Michela Verni, Carlo Giuseppe Rizzello

**Affiliations:** 1Department of Soil, Plant and Food Science, University of Bari Aldo Moro, 70126 Bari, Italy; chiara.demarinis@uniba.it (C.D.); marco.montemurro@ispa.cnr.it (M.M.); erica.pontonio@uniba.it (E.P.); 2National Research Council of Italy, Institute of Sciences of Food Production (CNR-ISPA), 70126 Bari, Italy; 3Department of Environmental Biology, “Sapienza” University of Rome, 00185 Rome, Italy; andrea.torreggiani@uniroma1.it (A.T.); carlogiuseppe.rizzello@uniroma1.it (C.G.R.)

**Keywords:** carob pulp, plant-based beverage, fermentation

## Abstract

Carob, an underutilized crop with several ecological and economic advantages, was traditionally used as animal feed and excluded from the human diet. Yet, nowadays, its beneficial effects on health are making it an interesting candidate as a food ingredient. In this study, a carob-based yogurt-like product was designed and fermented with six lactic acid bacteria strains, whose performances after fermentation and during shelf life were assessed through microbial and biochemical characterization. The strains showed different aptitudes to ferment the rice–carob matrix. Particularly, *Lactiplantibacillus plantarum* T6B10 was among the strains with the lowest latency phase and highest acidification at the end of fermentation. T6B10 also showed discrete proteolysis during storage, so free amino acids were up to 3-fold higher compared to the beverages fermented with the other strains. Overall, fermentation resulted in the inhibition of spoilage microorganisms, while an increase in yeasts was found in the chemically acidified control. The yogurt-like product was characterized by high-fiber and low-fat content; moreover, compared to the control, fermentation decreased the predicted glycemic index (−9%) and improved the sensory acceptability. Thus, this work demonstrated that the combination of carob flour and fermentation with selected lactic acid bacteria strains represents a sustainable and effective option to obtain safe and nutritious yogurt-like products.

## 1. Introduction

Global warming, environmental stresses, and other limitations call for a more careful consideration of minor or underutilized crops, and one of these valuable multifunctional, often neglected crops is carob (*Ceratonia siliqua* L.). Belonging to the Leguminosae or Fabaceae family, carob tree is a xerophytic species, resistant to droughts and salinity and well adapted to the Mediterranean region’s ecological conditions. Carob trees also have deep-root systems, which are intolerant to waterlogging and allow CO_2_ to sink, mitigating global warming effects. The ability to grow in warm areas with low chill requirements also implies the possible role of carob tree in fire protection of agroforest ecosystems [1]. Besides the ecological and economic advantages deriving from its cultivation, several beneficial effects on obesity, glycemia, dyslipidemia, meta-inflammation, and metabolic disorders have been bestowed upon carob pulp and seeds, mostly due to their polyphenols and fiber components [2,3]. Carob pods are rich in sugars (more than 30%), among which sucrose is the most abundant, and fibers (up to 40%) but poor in protein (2–7%). They also have appreciable amount of potassium, calcium, magnesium, phosphorous, and iron [4], while the seed endosperm also contains a high percentage of galactomannan, a polysaccharide widely utilized in the food industry due to its thickening, binding, and stabilizing properties, as well as its health benefits in lowering blood cholesterol levels, postprandial blood glucose, and insulin [5].

Traditionally, carob has been used to produce animal feed, and for many years, it was excluded from the human diet. Nowadays, the food industry’s diversification and innovative development are pushing to exploit carob as an ingredient in a variety of foods and beverages due to its cocoa-like aroma [3]. The potential of carob flour to be used as an ingredient in milk-based fermented beverages has been investigated mostly to increase fiber content [6]; however, concerns about the environmental impact and sustainability of animal-based diets, as well as human health issues thereof related, have fueled consumer demand for dairy alternatives, paving the way for plant-based yogurt-like products [7]. Consequentially, a wide range of novel beverages are constantly emerging. Few of them also include carob in the formulation (for a review see Vitali et al. [8]). Despite efforts to render such novel products similar to conventional ones in terms of appearance, the nutritional aspects of such yogurt-like products vary widely depending on the ingredients used. Frequently, plant-based alternatives present a high amount of sweetener to improve sensory characteristics and acceptance, and lower protein content compared to their counterparts [9]. Conversely, in favor of the plant-based ones, fiber and lipid contents tend to be, respectively, lower and higher in conventional yogurt [9]. 

Hence, based on the above considerations, in this study, a lactose- and gluten-free yogurt-like product (YL), was designed and six lactic acid bacteria (LAB) strains, isolated from several vegetable matrices, were used to ferment a rice–carob substrate. Rice flour was used to confer a creamy structure, thanks to the abundance of gelatinizable starch, and to soften the carob taste and flavor, which is commonly used in small amounts [8]. As for the LAB, *Lacticaseibacillus rhamnosus* SP1, a commercial probiotic strain, already employed for making quinoa [10] and oat flakes beverages [11], was chosen because of its already-demonstrated optimal technological properties and high survival under refrigerated storage conditions. *Lactiplantibacillus plantarum* T6B10, isolated from quinoa sourdough, showed the best adaptation and highest acidification when used as a starter in a quinoa YL [10]. *Weissella cibaria* P9, isolated from pineapple, and *Leuconostoc pseudomesenteroides* DSM20193 were chosen for their ability to produce exopolysaccharides (EPS) in several vegetable matrices [12,13], whereas *Levilactobacillus brevis* AM7 was selected due to its proteolytic activity and antimicrobial properties [14]. Lastly, *Enterococcus faecium* CA16, one of the few strains isolated from carob pulp flour, was also used due to its potential ability to adapt to the substrate. LAB performances after fermentation and during shelf life were assessed through microbial and biochemical characterization. A more in-depth nutritional and sensorial characterization was also performed on the YL fermented with the strain that showed the best aptitude.

## 2. Materials and Methods

### 2.1. Ingredients and Microbial Strains

Commercial rice (Molino Rivetti, Maclodio, Italy) and roasted carob (*Ceratonia siliqua* L.) pulp (Rapunzel Naturkost, Legau, Germany) flours were used in this study. The proximal composition (% *w*/*w*) of the flours was as follows for rice: moisture, 10.8%; total carbohydrates, 78.66% (of which sugars, 1.22%); dietary fibers, 0.45%; lipids, 2.14%; proteins, 8.21%; and salt, 0.01%. For carob pulp, the proximal composition was a follows: moisture, 6%; carbohydrates, 52% (of which sugars, 32%); dietary fiber, 37%; lipids, 0.5%; and protein, 5%.

Rice flour was classified as extra-fine with a particle size lower than 150 μm for approximately 90% and lower than 212 μm for approximately 10% of the total weight. 

Six lactic acid bacteria strains previously used for fermentation of vegetable matrices were used as starters: *Lp. plantarum* T6B10 [10], *Lv. brevis* AM7 [14], and *W. cibaria* P9 [12], belonged to the culture collection of the Department of Soil, Plant and Food Sciences (University of Bari, Italy); *Lc. rhamnosus* SP1 (also registered as DSM 21690) [10,11,15], a commercial probiotic strain provided by Sacco S.r.l. (Cadorago, Italy); and *Leuc. pseudomesenteroides* DSM20193 [16] and *E. faecium* CA16. 

In particular, *E. faecium* CA16 was isolated from carob pulp flour by picking colonies of presumptive LAB obtained from the plate count in De Man, Rogosa and Sharpe (MRS, Oxoid, Basingstoke, UK) of the 10^−3^ dilution of the sample. After assessing that the isolate was Gram-positive, catalase-negative, and non-motile it was re-streaked onto MRS agar medium after being cultured in MRS broth for 24 h at 30 °C. Genomic DNA from the pure culture was extracted using a Bacterial Genomic DNA Isolation Kit (Norgen Biotek Corp., Thorold, ON, Canada), according to the manufacturer’s instructions. To identify presumptive LAB, partial sequencing using primer LpigF/LpigR (5′-TACGGGAGGCAGCAGTAG-3′ and 5′-CATGGTGTGACGGGCGGT-3′) was used to amplify the 16S rRNA and compare the sequence obtained with those reported in the BLAST database [17].

Strains were routinely propagated in MRS at 30 °C for 24 h according to the isolation media and cultivation conditions described in the respective papers and reported elsewhere [10]. When used as starters for fermentation, LAB were cultivated until the late exponential phase of growth was reached (circa 16 h), harvested by centrifugation at 9000× *g* at 4 °C for 10 min, washed twice in 50 mM phosphate buffer (4 °C, pH 7.0), resuspended in tap water at a final cell density of circa 7 Log CFU (colony forming unit)/g, and inoculated in the yogurt-like products.

### 2.2. Biotechnological Protocol for YL Production

Rice and carob pulp fours were resuspended in tap water at 15 and 7% (*w*/*v*), respectively. The suspension was homogenized with an Oster 6805 (Jarden Consumer Solutions Ltd., Cheadle, UK) mixer and then subjected to a gelatinization process at 80 °C for 15 min as described by Montemurro et al. [13]. Then, the gelatinized mixture (100 mL for each replicate) was cooled at 4 °C until reaching a temperature of 30 °C prior to the inoculum of the starters (t0). The initial cell density of the inoculated strain was circa 7.0 Log CFU/mL. Fermentation was carried out at 30 °C for 16 h (Figure 1).

At the end of fermentation (tf), the YL was cooled down to 4 °C in 5 min and analyzed within 2 h after fermentation, and after 15 (t15) and 30 (t30) days of storage under refrigerated conditions. YL samples were coded as follows: CA16, 20193, SP1, T6B10, AM7, P9 (fermented with *E. faecium* CA16, *Leuc. pseudomesenteroides* DSM20193, *Lc. rhamnosus* SP1, *Lp. plantarum* T6B10, *Lv. brevis* AM7, and *W. cibaria* P9, respectively). A control sample (Ct) corresponding to a not inoculated but chemically acidified YL (corrected at pH 4.50 with lactic acid addition) was incubated in the same conditions of the inoculated YL and characterized.

### 2.3. YL Characterization

#### 2.3.1. Biochemical and Microbiological Characterization

The pH of the YL was determined with a pH meter M.507 (Crison, Milan, Italy) equipped with a food penetration probe, whereas total titratable acidity (TTA) was determined on 10 g of product homogenized with 90 mL of distilled water and expressed as a quantity (mL) of 0.1 M NaOH needed to reach a pH of 8.3.

During the fermentation process, pH was monitored every two hours. The kinetics of acidification were modeled according to the Gompertz equation as modified by Zwietering [18]: y = k + A exp {−exp[(Vmax^e^/A) (λ − t) + 1]}, where y is the acidification extent expressed as dpH at the time t; k is the initial level of the depend variable to be modeled; A is the pH difference between inoculation and the stationary phase; Vmax is the maximum acidification rate; and λ is the length of the latency phase expressed in hours.

YLs were characterized for cell densities of LAB, yeasts, molds, and enterobacteria. LAB were determined on MRS (Oxoid), supplemented with cycloheximide (0.1 g/L) incubating the plates, in anaerobic conditions at 30 °C for 48 h. Yeasts were plated on Sabouraud Dextrose Agar (SDA, Oxoid), supplemented with chloramphenicol (0.1 g/L), in aerobic conditions, at 25 °C for 48 h. Molds were enumerated on Potato Dextrose Agar (PDA, Oxoid) in aerobic conditions at 25 °C for 48 h. Total enterobacteria were determined on Violet Red Bile Glucose Agar (VRBGA, Oxoid) at 37 °C for 24 h. According to the Oxoid manual, after the medium was solidified with the sample inoculum, 10 mL of the same medium was overlayed and left to solidify.

Tris-HCl extracts (water soluble extracts, WSE) were prepared according to Weiss et al. [19] and employed for analyses of lactic and acetic acids and free amino acids (TFAAs). The kits K-DLATE and K-ACET (Megazyme, Bray, Ireland) were used for the determination of lactic and acetic acid concentrations, following the manufacturer’s instructions. TFAAs were analyzed using a Biochrom 30+ series Amino Acid Analyzer (Biochrom Ltd., Cambridge Science Park, UK) with a Li-cation-exchange column (20 by 0.46 cm inner diameter) [20].

Glucose, fructose, maltose, and sucrose were determined using the K-FRUGL and K-MASUG kits (Megazyme, Bray, Ireland), respectively.

#### 2.3.2. Technological Characterization

The water-holding capacity (WHC) was determined as described by Kovalenko and Briggs [21]. YL samples (10 g) were centrifuged at 1330× *g* for 5 min at room temperature. Supernatant fluid was drained for 1 min and the water-holding capacity was calculated as: [(W_sample_ − W_supernatant_)/W_sample_] × 100(1)

The viscosity was measured on 100 mL samples using the Viscotech Myr VR 3000 rotational viscometer equipped with the L4 probe (TQC, Capelle aan den ljssel, The Netherlands) at 100 rpm.

#### 2.3.3. Functional Characterization

The antioxidant potential of the YL was assessed through the determination of 2,2-diphenyl-1-picrylhydrazyl (DPPH) radical scavenging activity on the methanolic extract (ME) of the samples as described by Yu et al. [22]. Five grams of each sample was mixed with 50 mL of 80% methanol to obtain ME. The mixture was purged with nitrogen stream for 30 min, under stirring conditions, and centrifuged at 4600× *g* for 20 min. ME was transferred into test tubes, purged with nitrogen stream, and stored at ca. 4 °C before analysis. The scavenging activity was expressed as follows: DPPH scavenging activity (%) = [(blank absorbance − sample absorbance)/blank absorbance] × 100 (after 10 min of reaction). The value of absorbance was compared with 75 ppm butylated hydroxytoluene (BHT), which was used as the antioxidant reference.

### 2.4. Shelf-Life Monitoring

Aiming to investigate the YL quality, microbial and biochemical analysis were conducted after 15 and 30 days of refrigerated storage. More specifically, LAB, yeasts, molds, and enterobacteria cell densities were evaluated to assess the survival of the inoculated strains and of possible contaminants under refrigerated conditions. pH, TTA, organic acids, sugars, TFAAs, WHC, and viscosity were also determined as above described.

### 2.5. Selection and Characterization of a Rice–Carob YL

Based on the pro-technological, biochemical, and functional properties of the YL, when produced and during storage, *L. plantarum* T6B10 was selected as the most promising starter for fermentation. Hence, YL-T6B10 was subjected to further nutritional characterization.

#### 2.5.1. Nutritional Characterization and Starch Hydrolysis Index

Proximate composition of the YL was calculated using the following methods: ISO 2171: 2007, ISO 712: 2010, and ISO 16634: 2016 (part 2) for ash, moisture, and protein contents, respectively; the AOAC 985.29 method was used to measure total fiber; the fat content was assessed using the procedure outlined in the Italian D.M. n. 4 of 23 July 1994, whereas Legislative Decree No. 77 of 16 February 1993 was used to calculate carbohydrates from the differences in nutrients.

Starch hydrolysis index of YLs (HI) was determined by mimicking the in vivo digestion of starch [23]. YL aliquots, containing 1 g of starch, were subjected to a sequential enzymatic treatment, including an oral phase, gastric phase, and intestinal phase, using salivary α-amylase, pepsin, and pancreatic α-amylase, respectively [23]. The released glucose concentration was determined with the D-glucose assay Kit (GOPOD-format, Megazyme) following the manufacturer’s instructions. The degree of starch digestion was expressed as the percentage of potentially available starch hydrolyzed after 180 min. A control wheat bread (C-WB) was used as the reference to estimate the hydrolysis index (HI = 100). The equation pGI = 0.549 × HI + 39.71 proposed by Capriles and Arêas [24] was used to calculate the predicted glycemic index (pGI) 

#### 2.5.2. Sensory Analyses

Sensory analyses of the Ct and T6B10 YL (Tf) were carried out by 10 trained panelists (5 men and 5 women; average age: 31 years, range: 24–41 years) with demonstrated abilities and prior expertise in cereal-based product assessment. A two-hour training session was performed, and the assessors evaluated the descriptors to be included in the sessions. In particular, the following descriptors were chosen: color intensity (Cl), uniformity (Uf), adherence to spoon (Ad), and presence of particles for the appearance (Pr); overall odor intensity (Od), pungent smell (Pn.s), cocoa (Cc.s), and creamy (Cr.s) smell for the odor; sweet (Sw.t), salty (St.t), bitter (Bt.t), and acidic (Ac.t) for the taste; sweet (Sw.at), astringent (As.at), and earthy (Er.at) for the aftertaste evaluation.

The sensory assessments were conducted at the Department of Soil, Plant, and Food Science at the University of Bari, Italy, as previously described by Elia [25]. Three separate sessions were conducted, each including the assessment of the samples. The YLs were served in a random order and encoded with three-digit random numbers. These were rated on a scale from 0 to 10 in a sensory evaluation questionnaire, with 10 being the highest perception of the descriptor.

### 2.6. Statistical Analysis

All the microbiological, chemical, biochemical, textural, and sensory analyses were carried out in triplicate for each batch of samples. One-way analysis of variance (ANOVA) (*p* < 0.05), followed by Bonferroni’s post-hoc test, was performed using statistical software Statistica 12.5 (StatSoft Inc., Tulsa, OK, USA).

## 3. Results

### 3.1. YL Characterization

#### 3.1.1. Fermentation, Biochemical, and Functional Characterization

The pH of the rice–carob mixture employed as substrate for fermentation ranged from 5.5 to 5.6 (Table 1). The values observed after 16 h of fermentation were all lower than 5.0, but only T6B10 had a pH lower than 4.5 (4.33, Table 1), although the pH of 20193 did not significantly (*p* < 0.00238, Bonferroni-corrected) differ from the latter. SP1 and AM7 had a similar pH (4.71 and 4.68, respectively), while the highest values were reached in CA16 and P9. Accordingly, TTA values were the lowest for CA16 and P9, while the highest values were found in 20193 and T6B10 (Table 1).

All the strains employed as starters for fermentation were inoculated at cell densities ranging from 7.19 to 7.63 Log CFU/mL (Table 2). Despite the thermal treatment, the rice–carob gelatinized mixture used as substrate for fermentation still contained low *Enterobacteriaceae* (2.33–2.71 Log CFU/mL), yeasts (2.33–2.71 Log CFU/mL), and molds (2.00–2.28 Log CFU/mL) cell densities (Table 2). During incubation, all the inoculated strains increased by circa 2 Log cycles, with cell densities ranging from 9.00 (CA16) to 9.73 (T6B10), at tf. Moreover, *Enterobacteriacee* decreased significantly (*p* < 0.005, Bonferroni-corrected) in the chemically acidified (not inoculated) control (Ct) and in 20193, SP1, T6B10, AM7 and P9 compared to t0, while a similar density was found in CA16 and P9, both characterized by the highest values of pH among the different YLs (Table 2). Ct was also characterized by the highest number of yeasts and molds, although at values lower than 3.00 Log CFU/mL. No mold was detected in the inoculated samples at tf.

The kinetic of acidification for each strain was also monitored during incubation (Table 3). The lowest latency phase was observed for 20193, SP1, and T6B10 with a mean value of 3.53 h, whereas CA16 and P9 had the highest λ. T6B10 and 20193 were also the strains that caused the highest ΔpH and Vmax, respectively. 

The lactic acid concentration in YLs after fermentation ranged from 7.21 to 8.80 (CA16 and P9) to 20.17 mmol/L (T6B10). Intermediate values (11.69–13.98 mmol/L) were observed for 20193, SP1, and AM7 (Table 1). Low concentrations of acetic acid were present at the end of fermentation in all the inoculated samples; in particular, only in P9 and 20193 it was higher than 1.20 mmol/L.

Hexoses and disaccharides were also analyzed in the YLs before and after fermentation. Significant (*p* < 0.005, Bonferroni-corrected) decreases were found in glucose, whose final (tf) concentration was 50–70% lower than that observed at t0 for all inoculated samples, with the exception of CA16, in which only a decrease of 40% was observed (Table 1). Sucrose slightly, yet significantly (*p* < 0.005, Bonferroni-corrected), decreased only in 20193, SP1, and T6B10, during incubation. No significant variation was observed in maltose concentration during incubation, compared to t0, nor among the starters, while slight increases in fructose concentration were found in the inoculated samples (Table 1), especially in 20193 and T6B10 (+30% compared to the t0 concentration).

A concentration ranging from 159 to 163 mg/L of TFAAs characterized the YL substrate before fermentation (Table 1), and significant (*p* < 0.005, Bonferroni-corrected) decreases occurred during fermentation in CA16 (−52%), T6B10 (−29%), and 20,193 (−29%) and AM7 (−23%), while no significant (*p* > 0.005, Bonferroni-corrected) decreases were found in SP1 and P9.

In detail (Figure 2), aspartic acid (Asp), cysteine (Cys), proline (Pro), and valine (Val) were the most abundant FAAs in the YL substrate at t0. The activity of the LAB caused different changes in FAA profiles. Overall, threonine (Thr), serine (Ser), asparagine (Asn), glutamic acid (Glu), glycine (Gly), leucine (Leu), and arginine (Arg) significantly (*p* < 0.005) decreased in all the inoculated samples during fermentation, whereas Gly, alanine (Ala), tyrosine (Tyr), phenylalanine (Phe), lysine (Lys), histidine (His), and tryptophan (Trp), already found at low concentrations in t0, almost completely disappeared in all the fermented samples (Figure 2). Different balances characterized Asp, which completely disappeared in CA16, while it increased in SP1 and P9; Val, which decreased in AM7 by 90% and in all the other YL from 20 to 43%; and Pro, which increased (circa 50%) in CA16, SP1, AM7, and P9 but did not vary (*p* > 0.005) in 20193 and T6B10 (Figure 2).

The antioxidant activity of the YLs was calculated as the radical scavenging activity on DPPH radical of the methanolic extract of the samples. Intense activity was already found at t0, and fermentation did not affect the values, which were in any case higher than 80% (Table 1).

#### 3.1.2. Technological Characterization

WHC and viscosity of the YL were also determined. In detail, significant (*p* < 0.005, Bonferroni-corrected) increases in WHC were found for three of the inoculated samples (20193, T6B10, and SP1), compared to their respective t0 samples. Meanwhile, CA16, AM7, P9, and the chemically acidified control (Ct) did not show differences compared to the corresponding t0 samples (Table 1).

Viscosity data seem to be correlated to the acidification that occurred during fermentation; indeed, only CA16 and P9, characterized by the highest pH values at tf, were characterized by values higher than 12,000 mPa × s, while decreases of circa 30% were found in all the other samples compared to corresponding t0 samples (Table 1). 

### 3.2. Shelf-Life Monitoring and Starter Survival

During the first 15 days of refrigerated storage, the pH of all the YLs further decreased (0.2–0.7 units, Table 1).

Overall, values were lower than 4.5, with the lowest values observed for T6B10 and AM7 (3.94 and 3.97, respectively).

At the end of the 15 days of storage, LAB densities in all the inoculated samples were higher than 9.20 Log CFU/mL, without significant (*p* > 0.005, Bonferroni-corrected) differences with the corresponding samples at tf. No increases in *Enterobacteriaceae*, yeast, and molds were observed, except for Ct, in which indigenous LAB and molds were at a density of ca. 3.00 Log CFU/mL (Table 2).

Lactic and acetic concentrations increased in all the YLs. As expected, small increases in organic acids were found in Ct, as the results of the activity of microorganisms survived to the thermal treatment of the YL substrate. At 15 and 30 days of storage, the highest content of lactic acid was found in AM7 and T6B10, while acetic acid was at a concentration higher than 5 mmol/L in T6B10, P9, and 20193. Among the sugars analyzed, the most consistent decreases (up to 3 g/L) in inoculated YL samples were found for sucrose. In comparison to tf, very slight decreases were found in fructose concentrations at t15, except for CA16, in which the lowest final content was found. During storage, TFAA concentration fluctuated depending on the strain used for fermentation; in particular, compared to t0, it increased significantly (*p* < 0.005, Bonferroni-corrected) in T6B10 (+41%) and decreased by 20% in AM7.

SP1, T6B10, and P9 were characterized by the highest TFAA concentration; in particular, Asp, Val, and Pro were the amino acids present at the highest concentration. Among these three YL samples, T6B10 was characterized by the highest content in Gly, Tyr, Phe, Lys, Trp, and Arg (Figure 3A).

Slight decreases were found in WHC in all the samples. Conversely, the viscosity values were strongly affected by the long storage time at 4 °C. In particular, a decrease of 40% was found for the not inoculated control (Ct), which was characterized by a significantly (*p* < 0.00238, Bonferroni-corrected) lower value compared to the inoculated samples. The highest viscosity at t15 was observed for 20193 and CA16, followed by SP1 and T6B10 (Table 1).

The radical scavenging activity was always higher than 83%, without significant differences among the samples.

At the end of the 30 days of refrigerated storage, LAB cell density in YLs decreased by ca. 1 Log CFU/mL. Only CA16 was characterized by LAB lower than 8.00 Log CFU/mL (Table 2). Yeasts persisted in all the inoculated YL samples, while a significant increase was observed in Ct. At 30 days, no *Enterobacteriaceae* or molds were found in any samples (Table 2).

Moreover, a slightly but significant decrease in pH was observed for all the samples compared to corresponding t15, together with increases in lactic and acetic acids (Table 1). A further decrease in sucrose also occurred. Regarding the TFAAs, a significant increase characterized T6B10 (+87 and +46% compared to tf and t15, respectively). A significant (*p* < 0.005, Bonferroni-corrected) increase in TFAAs was also observed in 20193 (+41% compared to t15), although the final concentration was markedly lower than that found for T6B10 (Table 1).

Besides the abundance of Asp, Cys, and Pro (Figure 3B), T6B10 at t30 was characterized by concentrations of Thr, Ser, Glu, Val, Met, Ile, leu, Tyr, Phe, GABA, Orn, Lys, His, Trp, and Arg higher than the other inoculated samples.

### 3.3. Characterization of YL-T6B10

Based on these results, *L. plantarum* T6B10 was the starter capable of the most intense and fast acidification of the substrate. Moreover, YL-T6B10 was characterized by the highest concentration and better-balanced mixture of FAAs. For these reasons, YL-T6B10 was further characterized for the nutritional and sensory profiles.

#### 3.3.1. Nutritional Characterization and Predicted Glycemic Index

YL-T6B10 was characterized by the following nutritional label: proteins, 1.58 g/100 g; fats, 0.36 g/100 g; carbohydrates, 15.41 g/100 g (of which sugars, 2.40); and 2.66 g/100 g dietary fibers. The energy value was 76.34 Kcal/100 g. 

The starch hydrolysis indexes (HI) of the two samples analyzed were 53.47 ± 3.45 and 37.99 ± 1.01%, respectively, for Ct and T6B10. Consequently, the pGI of Ct was markedly and significantly (*p* < 0.05) higher than that of T6B10 (69.07 ± 1.89 vs. 60.57 ± 0.91).

#### 3.3.2. Sensory Analysis

Aiming to describe the sensory profile of YL-T6B10 and highlight the effect of fermentation (in comparison to a chemically acidified substrate), a list of descriptors was selected during the preliminary sessions of analysis. The descriptor list includes attributes for the appearance, smell, taste, and aftertaste of the products, many of which were chosen because they were peculiar compared to conventional milk yogurt (adherence to spoon, acidic taste, creamy smell) [24] or to plant-based products (astringent taste and aftertaste, earthy taste) [24] or characteristic of carob-derived ingredients (cocoa smell) [25]. YL-T6B10 appearance was characterized by a very high score for uniformity and adherence to spoon (Figure 4). For both attributes, values were significantly higher than the control. Odor intensity and cocoa smell were also intensified by fermentation. Sweet and bitter taste perceptions were lower in fermented samples compared to the control, and an astringent and earthy aftertaste was significantly lower in YL-T6B10 (Figure 4).

## 4. Discussion

Carob is a multifunctional valuable and underutilized crop, whose cultivation can lead to several ecological and economic advantages [1,2]. Its consumption was found to have numerous beneficial effects on human health, mainly due to its polyphenol and fiber content [2,3]. It was indeed suggested that the development of fermented beverages from this crop can contribute to the Sustainable Development Goals by promoting human health and sustainable production and consumption [8]. Moreover, the global plant-based beverage market has increased in the last decade, reaching a value of almost USD 20,000 million in 2023 [26], and several research papers have explored the possibility of using carob flour as an ingredient in new health-promoting beverages [7,8].

Hence, in this study, a lactose- and gluten-free yogurt-like product, containing a mixture of rice and carob flour, was designed. The experimental YL developed could be considered “high-fiber” because it contains more than 3 g of fiber per 100 kcal, according to the European Regulation (EC) No 1924/2006. However, its protein content was not high enough to reach the “source of protein” claim, hence further studies could focus on recipe formulation. When lacking in the ingredients used for their formulation, plant-based beverages are often fortified with micronutrients such as calcium, yet the bioavailability of the fortifier, more than its amount, should be considered [9]. In this regard, fermentation with lactic acid bacteria is known, among other things, to increase the bioaccessibility of nutrients [27]. Reports on the use of fermentation to increase the bioavailability of carob nutrients while decreasing possible anti-nutritional factors can already be found [8]. The high presence of simple carbohydrates also makes carob an optimal substrate for fermentation by microorganisms. Furthermore, lactic acid bacteria also play a pivotal role in the sensory aspects of fermented food because carbohydrate metabolism, as well as amino acid catabolism, provide acids, aldehydes, ketones, and many other compounds that can influence flavor and taste [28]. Fermentation directly or indirectly affects food nutritional features of food by reducing anti-nutritional compounds, increasing protein digestibility, or decreasing the predicted glycemic index [27,29]. However, most of all, lactic acid bacteria fermentation is used to extend the shelf life or promote food probiotic potential [28,30]. Hence, the yogurt-like product developed in this study was fermented, yet when optimizing a fermentation process, starter selection plays a key role, and the need for proper LAB starters of alternative vegetable matrixes has been largely recognized [30,31]. In the food industry, several microbial cultures are used as starters to control the fermentation process, reducing the risk of fermentation failure while ensuring safety, stability, and suitable sensory profiles. The starter selection includes a systematic approach, which progressively reduces the number of candidates and involves (i) the assessment of the ability to adapt to the raw material resisting stress conditions; (ii) the identification of key metabolite producers; and (iii) the evaluation of technological parameters [31]. The strains used (*Lp. plantarum* T6B10, *Lv. brevis* AM7, *W. cibaria* P9, *Lc. rhamnosus* SP1, and *Leuc. pseudomesenteroides* DSM20193) were selected for the above-mentioned features and optimal performances in the fermentation of other YLs [10,11,12,13,14], whereas *E. faecium* CA16 was selected because it was one of the few isolated from carob. Nevertheless, the possibility to be used as starters is still debated because bacteria belonging to the genus *Enterococcus* are frequent opportunistic human pathogens and express adhesion factors; thus, they do not belong to the QPS (qualified presumption of safety) list, yet some strains are used as starter cultures in dairy products. Hence, a deep level of investigation in a case-by-case assessment should be provided before using such microorganisms as starters at the industrial level [32].

Despite being isolated from carob and, presumably, the one that might have exhibited higher resistance to the matrix condition, CA16, along with P9, was the strain that worst adapted to the matrix (Table 2). Compared to the other strains, they were the ones with the longest latency phase, determining slow and low acidification, corroborated by the low organic acid content found at tf, thus resulting in a favorable environment for *Enterobacteriaceae*. Similar to milk-based yogurt, fermented plant-based YLs normally have a pH lower than 4.5, which is conventionally considered a value for product microbial safety and decreases during storage because of post-acidification by LAB [27]. The importance of low pH in the pathogenesis of enteric bacteria is well known, which is why rapid acidification is fundamental to guaranteeing the hygienic safety of fermented food. Indeed, when entering the cell membrane, organic acids dissociate, resulting in the acidification of the cytoplasm, and the continuous influx of protons is responsible for depleting cellular energy, as well as cell death [33]. Nevertheless, further acidification was obtained during the refrigerated storage of CA16 and P9, resulting in the inhibition of spoilage microorganisms. The same could not be said for the control: although chemical acidification prevented the growth of molds and bacteria that withstood thermal treatment, compared to fermented YL and Ct before fermentation, an increase of more than two log cycles in yeast cell density was observed during the storage. Hence, even though chemical acidification can be considered suitable for ensuring the microbiological stability of the YL, the upside of fermentation and consequent LAB competition for nutrients cannot be overlooked. All the other strains better adapted to the matrix conditions, as showed by the shorter latency phase, as well as the higher speed of acidification. Being the only obligate heterofermentative strains used [34], DSM20193 and P9 produced the highest amount of acetic acid (Table 1). Consequentially, because lactate is the main product of metabolism when fermentable carbohydrates are abundant [35], organic acid production, directly related to carbohydrate metabolism, determined a decrease in glucose. Although all the strains are able to ferment the hexoses in the matrix [36], glucose is the preferred source of carbon for homolactic metabolism (as in the case of *E. faecium* CA16), whereas fructose, sucrose, and/or maltose are preferred during heterolactic metabolism (e.g., *Lv. brevis* AM7, *W. cibaria* P9, and *Leuc. pseudomesenteroides* DSM20193) [35]. In DSM20193, however, a significant increase in fructose was found after fermentation compared to the other samples. Indeed, fructose accumulation, when fermenting with EPS-producing strains, is common because in the presence of sucrose, the strains activate the production of dextransucrase, which releases fructose [36]. Indeed, sucrose concentration in DSM20193 decreased after fermentation. 

Overall, in the fermented YL, intense proteolysis was not observed, most likely due to the thermal treatment, which inactivated endogenous enzymes, thus limiting the synthesis of new amino acids so that in most of the strains, FAA content decreased after fermentation. Furthermore, low pH tends to shift lactic metabolism from hexose fermentation to amino acids utilization so that glutamate-, histidine- and tyrosine-decarboxylases play a major role in pH homeostasis and the stationary phase survival of LAB [36]. A similar result was obtained when an oat flake [11] and a hemp-based [13] YLs, both subjected to gelatinization, were fermented with LAB starters. Moreover, it has already been shown that protease activity is low during the initial fermentation stage because LAB utilize free amino acids already present in the matrix [35]. Indeed, in this study, fermentation lasted only 16 h. Nevertheless, T6B10 was the strain that determined the highest and more balanced FAA content after fermentation, which increased during the storage period as pH decreased below 4 (Table 1). 

In conventional yogurt, acidification directly impacts the stability of casein micelles, thus leading to the classic texture that characterizes them. Meanwhile, in plant-based YLs, optimal rheology is usually achieved with additives (protein extracts, inulin, thickeners, and emulsifiers) that do not fit the concept of “clean label” products [37,38]. Although a clear definition of “clean label” products does not exist, this concept highly depends on consumer perceptions, and labels such as “free from allergens”, “no GMOs”, “without artificial additives”, “minimally processed”, “simple/short ingredient lists”, and “transparent packaging” are all symbols of food product cleanness for consumers [37]. For this reason, a gelatinization step was included to increase the viscosity of the YLs, thus preventing phase separation and decreasing the entity of endogenous microbial contamination before starter inoculum; however, a decreasing trend during the storage was reported. Surely, due to the acidification and proteolysis, which slowly continued during storage, the instability of the plant protein structure might have resulted in a weakening of the product structure and a separation of the aqueous phase [39], leading to the lower WHC and viscosity observed (Table 1). For this reason, aiming to discriminate the effect of pH from potential metabolites produced, the YL used as a control was chemically acidified, thus confirming the impact of pH. Nevertheless, because EPS-like dextran, produced by the enzyme dextransucrase, has hydrocolloidal properties [38], EPS-producer strains DSM20193 and P9 were also chosen, and the YL viscosity was investigated. Although not as much as expected, compared to Ct, an increase in viscosity (up to 56%) was observed, mostly in P9 and CA16; overall, fermentation provided better rheology during storage. 

Being the strain that presented the best aptitude for fermentation for most of the parameters considered, T6B10 was chosen for further characterization. Indeed, compared to Ct, fermentation led to lower starch hydrolysis index and predicted glycemic index, which estimate how the amount of carbohydrates in foods raises blood glucose levels. The explanation of this phenomenon is not fully clear, but several aspects, among which are biological acidification, increase in the amount of resistant starch and soluble fibers, fast gastric emptying, and stimulation of satiety hormones, are considered the main mechanisms behind such a result [29]. Besides improving its nutritional quality, fermentation also improved the overall sensory acceptability of the YL, conferring a pleasant aroma with notes of fermented cocoa. The typical cocoa-like aroma is mostly due to carob flour, which is often regarded as a cheaper and healthier substitute of cocoa because of the lower fat content [3]; however, in this case, it was heightened by fermentation. Indeed, microbial catabolism of amino acids leads to the generation of several compounds useful for flavor formation. For example, Asp, one of the most abundant FAAs in the YL, is converted into compounds contributing to the buttery flavors [28], thus providing a scent that might remind consumers of common yogurt. Meanwhile Tyr, Phe, and Trp, which decreased during fermentation, are precursors of compounds that can confer flowers and bitter almond flavors [28] and, as a result, higher odor intensity and lower bitter and astringency taste were found for T6B10 YL compared to the control. 

## 5. Conclusions

This work demonstrated that carob pulp powder, which represents a by-product of locust bean gum production from carob seeds, can be used as an ingredient in the formulation of plant-based beverages, complying with the current concept of sustainability. The carob-based YL designed in this study was characterized by high-fiber and low-fat content for a product category that often presents an unbalanced nutritional profile. Moreover, fermentation with selected LAB strains represented a sustainable and effective option to exploit its potential, guaranteeing microbial safety, while improving the overall nutritional and sensory profile.

## Figures and Tables

**Figure 1 microorganisms-11-01607-f001:**
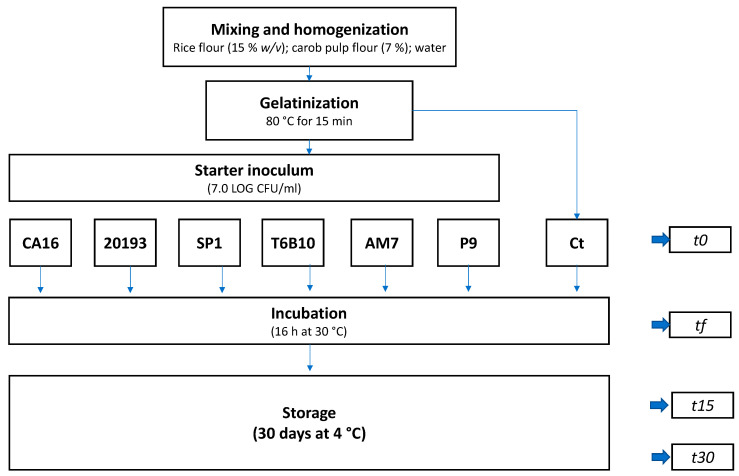
Flow chart of the YL-making process. YLs were fermented with *Enterococcus faecium* CA16, *Leuconostoc pseudomesenteroides* DSM20193, *Lacticaseibacillus rhamnosus* SP1, *Lactiplantibacillus plantarum* T6B10, *Levilactobacillus brevis* AM7, and *Weissella cibaria* P9 (CA16, 20193, SP1, T6B10, AM7, P9 are the respective YLs) at 30 °C for 16 h. Ct: incubated but not inoculated control.

**Figure 2 microorganisms-11-01607-f002:**
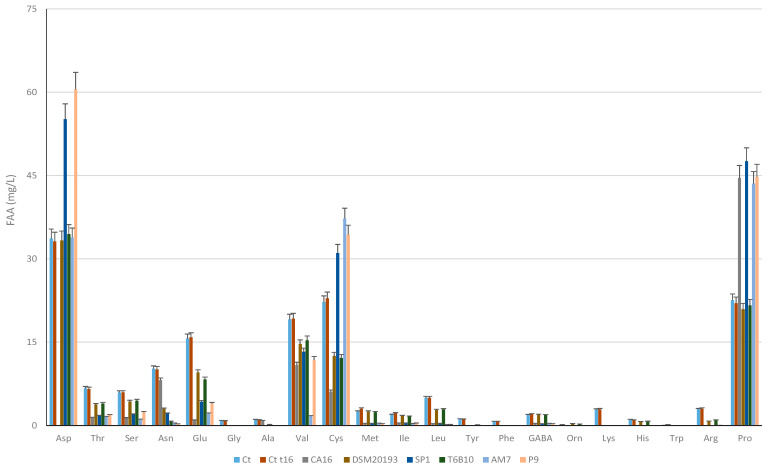
Free amino acids concentrations (mg/L) in YLs before (t0) and after incubation at 30 °C for 16 h (tf). CA16, 20193, SP1, T6B10, AM7, P9 are the YLs fermented with *Enterococcus faecium* CA16, *Leuconostoc pseudomesenteroides* DSM20193, *Lactobacillus rhamnosus* SP1, *Lactiplantibacillus plantarum* T6B10, *Levilactobacillus brevis* AM7, and *Weissella cibaria* P9, respectively. A control sample (Ct) corresponding to a not inoculated, but chemically acidified, YL was also characterized. The data are the means of three independent experiments ± standard deviations (n = 3).

**Figure 3 microorganisms-11-01607-f003:**
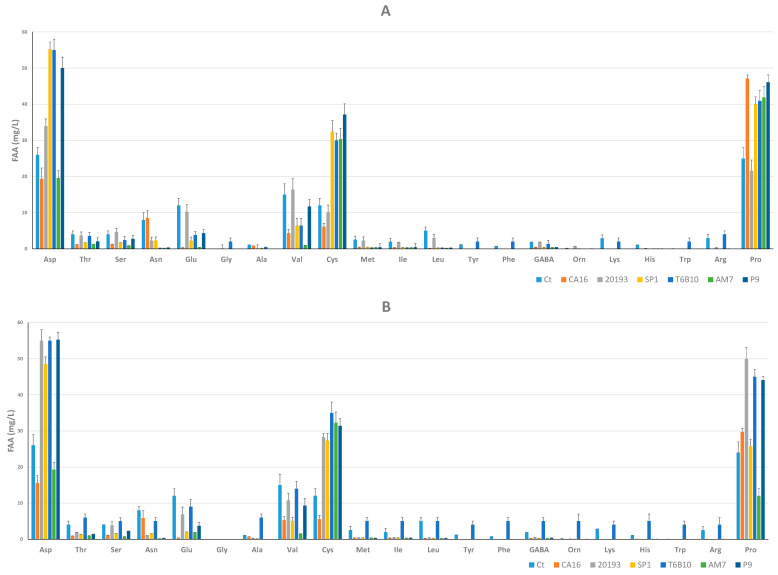
Free amino acids concentrations (mg/L) in YLs after 15 (t15, panel (**A**)) and 30 (t30, panel (**B**)) days of storage at 4 °C. CA16, 20193, SP1, T6B10, AM7, and P9 are the YLs fermented with *Enterococcus faecium* CA16, *Leuconostoc pseudomesenteroides* DSM20193, *Lactobacillus rhamnosus* SP1, *Lactiplantibacillus plantarum* T6B10, *Levilactobacillus brevis* AM7, and *Weissella cibaria* P9, respectively. A control sample (Ct) corresponding to a not inoculated, but chemically acidified, YL was also characterized. The data are the means of three independent experiments ± standard deviations (n = 3).

**Figure 4 microorganisms-11-01607-f004:**
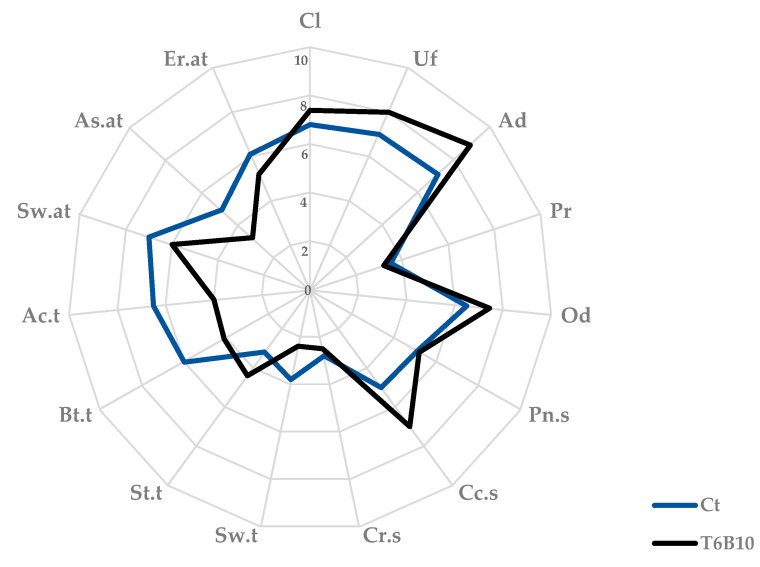
Sensory analysis of the yogurt-like T6B10, fermented with *Lactiplantibacillus plantarum* T6B10 at 30 °C for 16 h (tf). A control sample (Ct) corresponding to a not inoculated, but chemically acidified, YL incubated in the same conditions was included in the analysis. Descriptors: color intensity (Cl), uniformity (Uf), adherence to spoon (Ad), and presence of particles for the appearance (Pr); overall odor intensity (Od), pungent smell (Pn.s), cocoa (Cc.s), and creamy (Cr.s) smell for the odor; sweet (Sw.t), salty (St.t), bitter (Bt.t), and acidic (Ac.t) for the taste; sweet (Sw.at), astringent (As.at), and earthy (Er.at) for the aftertaste evaluation.

**Table 1 microorganisms-11-01607-t001:** Main characteristics of the YL before (t0) and after incubation at 30 °C for 16 h (tf) and after 15 (t15) and 30 (t30) days of storage at 4 °C. CA16, 20193, SP1, T6B10, AM7, P9 are the YLs fermented with *Enterococcus faecium* CA16, *Leuconostoc pseudomesenteroides* DSM20193, *Lacticaseibacillus rhamnosus* SP1, *Lactiplantibacillus plantarum* T6B10, *Levilactobacillus brevis* AM7, and *Weissella cibaria* P9, respectively. A control sample (Ct) corresponding to a not inoculated, but chemically acidified, YL was also characterized.

	pH	TTA	Lactic Acid (g/L)	Acetic Acid (mg/L)	Glucose (g/L)	Fructose (g/L)	Maltose (g/L)	Sucrose (g/L)	TFAA (mg/L)	WHC (%)	Viscosity (mPa × s)
** *t0* **											
**Ct**	4.50 ± 0.01 Ab	3.4 ± 0.3 BA	1.08 ± 0.01 Ca	n.d.	1.65 ± 0.01 Aa	2.16 ± 0.04 Aa	0.51 ± 0.03	16.99 ± 0.30 Aa	159 ± 6 Aa	84.44 ± 0.10 Aa	14,540 ± 36 Aa
**CA16**	5.50 ± 0.02 Aa	2.4 ± 0.2 Db	0.09 ± 0.00 Cb	n.d.	1.65 ± 0.05 Ba	2.15 ± 0.04 Aa	0.50 ± 0.03	17.02 ± 0.33 Aa	160 ± 4 Aa	84.22 ± 0.12 Aa	14,530 ± 27 Aa
**20193**	5.50 ± 0.03 Aa	2.4 ± 0.0 Cb	0.09 ± 0.01 Db	n.d.	1.65 ± 0.03 Aa	2.16 ± 0.03 Ba	0.52 ± 0.02	16.99 ± 0.29 Aa	163 ± 7 Aa	84.14 ± 0.10 Ba	14,535 ± 56 Aa
**SP1**	5.50 ± 0.04 Aa	2.4 ± 0.1 Db	0.09 ± 0.00 Cb	n.d.	1.65 ± 0.03 Ba	2.14 ± 0.04 Ba	0.50 a ± 0.03	16.92 ± 0.35 Aa	159 ± 6 Aa	84.40 ± 0.10 Ba	14,540 ± 47 Aa
**T6B10**	5.50 ± 0.02 Aa	2.4 ± 0.2 Cb	0.09 ± 0.01 Db	n.d.	1.65 ± 0.05 Aa	2.15 ± 0.03 Ca	0.50 ± 0.01	16.95 ± 0.34 Aa	158 ± 8 Ba	84.30 ± 0.10 Ba	14,542 ± 29 Aa
**AM7**	5.60 ± 0.03 Aa	2.2 ± 0.0 Db	0.09 ± 0.01 Cb	n.d.	1.65 ± 0.01 Ba	2.15 ± 0.07 Aa	0.52 ± 0.03	16.89 ± 0.34 Aa	160 ± 7 Aa	84.22 ± 0.11 Aa	14,534 ± 46 Aa
**P9**	5.60 ± 0.03 Aa	2.2 ± 0.1 Db	0.09 ± 0.00 Db	n.d.	1.65 ± 0.06 Aa	2.16 ± 0.04 Ba	0.52 ± 0.02	16.91 ± 0.31 Aa	160 ± 6 Aa	84.49 ± 0.12 Aa	14,538 ± 19 Aa
** *tf* **											
**Ct**	4.50 ± 0.04 Ac	3.4 ± 0.1 Bb	1.09 ± 0.03 Cb	n.d.	1.66 ± 0.02 Aa	2.14 ± 0.01 Ac	0.52 ± 0.03	16.86 ± 0.33 Aa	159 ± 6 Aa	82.26 ± 0.15 Bc	8180 ± 11 Bc
**CA16**	4.91 ± 0.03 Ba	3.0 ± 0.3 Cb	0.65 ± 0.01 Bc	12.6 ± 0.6 Bc	1.01 ± 0.04 Cb	2.41 ± 0.03 Ab	0.45 ± 0.05	17.04 ± 0.32 Aa	76 ± 4 Cc	85.82 ± 0.28 Ab	12,760 ± 72 Ba
**20193**	4.56 ± 0.06 Bc	4.0 ± 0.1 Ba	1.05 ± 0.08 Cb	76.9 ± 2.1 Ca	0.43 ± 0.01 Bd	2.79 ± 0.07 Aa	0.46 ± 0.04	16.06 ± 0.04 Bb	113 ± 3 Bb	93.88 ± 0.19 Aa	9920 ± 66 Bb
**SP1**	4.71 ± 0.02 Bb	3.6 ± 0.2 Cb	1.22 ± 0.04 Bb	39.0 ± 1.7 Bb	0.47 ± 0.02 Cd	2.53 ± 0.02 Ab	0.50 ± 0.03	16.02 ± 0.06 Bb	159 ± 8 Aa	90.59 ± 0.10 Aa	9720 ± 58 Bb
**T6B10**	4.33 ± 0.10 Bc	4.4 ± 0.2 Ba	1.46 ± 0.06 Ca	30.1 ± 2.0 Bb	0.54 ± 0.07 Cd	2.83 ± 0.06 Aa	0.56 ± 0.02	15.85 ± 0.09 Bb	112 ± 4 Cb	91.20 ± 0.18 Aa	9770 ± 45 Bb
**AM7**	4.68 ± 0.06 Bb	3.4 ± 0.0 Cb	1.26 ± 0.02 Bb	36.6 ± 1.1 Cb	0.80 ± 0.02 Cc	2.44 ± 0.18 Ab	0.50 ± 0.02	16.47 ± 0.26 Aa	123 ± 6 BCb	84.41 ± 0.28 Ab	9800 ± 63 Bb
**P9**	4.87 ± 0.05 Ba	3.0 ± 0.1 Cb	0.79 ± 0.05 Cc	81.7 ± 3.7 Ba	0.45 ± 0.03 Bd	2.57 ± 0.05 Ab	0.55 ± 0.01	16.84 ± 0.32 Aa	156 ± 7 Aa	84.16 ± 0.38 Ab	12,020 ± 159 Ba
** *t15* **											
**Ct**	4.32 ± 0.06 Ba	4.0 ± 0.2 Ab	1.39 ± 0.04 Bcd	44.4 ± 2.9 Be	1.53 ± 0.03 Ba	2.10 ± 0.02 Aa	n.d.	16.50 ± 0.43 Aa	128 ± 4 Bb	80.81 ± 0.45 Cb	3250 ± 27 Cd
**CA16**	4.39 ± 0.03 Ca	4.4 ± 0.6 Bb	1.71 ± 0.02 Ab	131.5 ± 10.3 Ad	0.61 ± 0.05 Db	1.01 ± 0.06 Bc	n.d.	15.09 ± 0.62 Ba	91 ± 3 Bc	80.60 ± 0.20 Cb	5810 ± 59 Ca
**20193**	4.33 ± 0.07 Ca	4.5 ± 0.3 Bb	1.52 ± 0.09 Ac	315.3 ± 11.2 Bb	0.25 ± 0.01 Cd	2.65 ± 0.13 Aa	n.d.	13.03 ± 0.08 Cc	113 ± 1 Bab	79.16 ± 0.13 Cb	5970 ± 67 Ca
**SP1**	4.42 ± 0.05 Ca	4.2 ± 0.1 Bb	1.63 ± 0.06 Ac	260.0 ± 7.9 Ac	0.56 ± 0.06 Cb	2.60 ± 0.10 Aa	n.d.	14.32 ± 0.75 Cb	144 ± 8 Aa	83.15 ± 0.28 Ca	5220 ± 133 Cb
**T6B10**	3.94 ± 0.03 Cb	5.9 ± 0.8 Aa	1.89 ± 0.01 Bb	362.7 ± 16.8 Aa	0.64 ± 0.07 Cb	2.23 ± 0.09 Cb	n.d.	13.57 ± 0.32 Cb	159 ± 6 Ba	80.26 ± 0.17 Cb	4970 ± 120 Cc
**AM7**	3.97 ± 0.01 Cb	5.9 ± 0.6 Ba	2.16 ± 0.08 Aa	138.1 ± 9.4 bd	0.50 ± 0.08 Db	2.09 ± 0.08 Abb	n.d.	15.62 ± 0.39 Ba	97 ± 12 Cc	78.92 ± 0.41 Bb	3710 ± 89 Cc
**P9**	4.32 ± 0.0 Ca	4.4 ± 0.1 Bb	1.40 ± 0.07 Bc	353.7 ± 5.6 Aa	0.40 ± 0.02 Bc	2.67 ± 0.04 Aa	n.d.	16.00 ± 0.57 Aba	156 ± 7 Aa	79.70 ± 0.25 Bb	3070 ± 206 Cd
** *t30* **											
**Ct**	4.15 ± 0.02 Ca	4.3 ± 0.1 Ac	1.56 ± 0.06 Ab	57.0 ± 2.8 Ad	1.51 ± 0.02 Bc	2.18 ± 0.05 Ab	n.d.	16.21 ± 0.39 Aa	126 ± 7 Bc	75.25 ± 0.05 Db	2840 ± 83 Cc
**CA16**	4.04 ± 0.07 Da	5.6 ± 0.3 Ab	1.78 ± 0.10 Ab	164.5 ± 8.9 Ac	2.37 ± 0.23 Ab	0.92 ± 0.07 Bd	n.d.	15.05 ± 0.45 Bab	66 ± 5 Cd	78.02 ± 0.15 Da	3970 ± 28 Da
**20193**	4.15 ± 0.01 Da	5.8 ± 0.2 Ab	1.62 ± 0.02 Ab	381.3 ± 24.3 Aa	0.57 ± 0.09 Bd	2.72 ± 0.14 Aa	n.d.	13.04 ± 0.20 CBc	160 ± 9 Ab	73.71 ± 0.24 Dc	3450 ± 56 Db
**SP1**	4.19 ± 0.05 Da	5.0 ± 0.5 Ab	1.7 ± 0.07 Ab	269.0 ± 10.2 Ab	3.05 ± 0.17 Aa	2.39 ± 0.21 Abb	n.d.	14.07 ± 0.41 Cb	115 ± 2 Bc	74.90 ± 0.23 Db	3340 ± 44 Db
**T6B10**	3.75 ± 0.05 Db	6.6 ± 0.4 Aa	2.32 ± 0.11 Aa	383.7 ± 18.8 Aa	1.17 ± 0.12 Bc	2.51 ± 0.02 Bd	n.d.	13.05 ± 0.14 Cc	231 ± 6 Aa	74.55 ± 0.26 Dbc	3540 ± 71 Db
**AM7**	3.76 ± 0.03 Db	6.8 ± 0.3 Aa	2.35 ± 0.13 Aa	168.2 ± 6.3 Ac	2.60 ± 0.18 Ab	1.95 ± 0.02 Bc	n.d.	15.17 ± 0.53 Bab	70 f ± 11 Cd	74.03 ± 0.23 Cc	2490 ± 39 Dd
**P9**	4.15 ± 0.08 Ca	5.7 ± 0.1 Ab	1.71 ± 0.08 Ab	396.3 ± 20.7 Aa	0.61 ± 0.07 Bd	2.22 ± 0.13 Bb	n.d.	15.79 ± 0.21 Ba	149 ± 3 Ab	73.25 ± 0.17 Cd	2750 ± 79 Cc

The data are the means of three independent experiments. Different uppercase letters between different incubation time in the same sample mean significant differences at a *p* value of <0.005 (*p*, Bonferroni-corrected). Different lowercase letters, among different sample within the same time of incubation mean significant differences at a *p* value of <0.00238 (*p*, Bonferroni-corrected). TFAAs: total free amino acids.

**Table 2 microorganisms-11-01607-t002:** Microbiological analysis of the YL before (t0) and after (tf) fermentation at 30 °C for 16 h and after 15 (t15) and 30 (t30) days of storage at 4 °C. CA16, 20193, SP1, T6B10, AM7, P9 are the YLs fermented with *Enterococcus faecium* CA16, *Leuconostoc pseudomesenteroides* DSM20193, *Lacticaseibacillus rhamnosus* SP1, *Lactiplantibacillus plantarum* T6B10, *Levilactobacillus brevis* AM7, and *Weissella cibaria* P9, respectively. A control sample (Ct) corresponding to a not inoculated, but chemically acidified, YL was also characterized.

	LAB (Log CFU/mL)	*Enterobacteriaceae* (Log CFU/mL)	Yeasts (Log CFU/mL)	Molds (Log CFU/mL)
	** *t0* **			
**Ct**	0.59 ± 0.03 Cc	2.53 ± 0.07 Aa	2.02 ± 0.02 Ca	2.28 ± 0.02 Aa
**CA16**	7.34 ± 0.27 Ba	2.62 ± 0.06 Aa	1.90 ± 0.02 Ba	2.02 ± 0.09 Aa
**20193**	7.63 ± 0.31 Ba	2.53 ± 0.07 Aa	2.10 ± 0.06 Ba	2.00 ± 0.09 Aa
**SP1**	7.19 ± 0.16 Cb	2.71 ± 0.07 Aa	2.10 ± 0.04 Aba	2.24 ± 0.01 Aa
**T6B10**	7.19 ± 0.23 Cab	2.52 ± 0.09 Aa	2.05 ± 0.07 Ba	2.10 ± 0.03 Aa
**AM7**	7.23 ± 0.32 Cab	2.33 ± 0.05 Aa	1.98 ± 0.08 Ba	2.05 ± 0.05 Aa
**P9**	7.33 ± 0.18 Cb	2.61 ± 0.04 Aa	2.04 ± 0.07 Ba	2.22 ± 0.05 Aa
	** *tf* **			
**Ct**	2.60 ± 0.13 B	1.72 ± 0.03 Bc	3.00 ± 0.13 Ba	2.10 ± 0.01 Aa
**CA16**	9.00 ± 0.52 A	2.52 ± 0.07 Aa	1.70 ± 0.05 Bc	<10 UFC/mL
**20193**	9.51 ± 0.47 Aa	1.83 ± 0.02 Bb	2.00 ± 0.04 Bb	<10 UFC/mL
**SP1**	9.62 ± 0.29 Aa	1.84 ± 0.02 Bb	2.00 ± 0.02 Bb	<10 UFC/mL
**T6B10**	9.73 ± 0.32 Aa	1.52 ± 0.07 Bc	2.10 ± 0.01 Bb	<10 UFC/mL
**AM7**	9.52 ± 0.63 Aa	2.01 ± 0.06 Bb	2.00 ± 0.06 Bb	<10 UFC/mL
**P9**	9.33 ± 0.38 Aa	2.32 ± 0.11 Bab	2.20 ± 0.05 Bb	<10 UFC/mL
	** *t15* **			
**Ct**	3.70 ± 0.31 Ab	1.02 ± 0.01 Cc	3.30 ± 0.14 Ba	<10 UFC/mL
**CA16**	9.20 ± 0.45 Aa	1.20 ± 0.02 Bb	2.30 ± 0.08 Ab	<10 UFC/mL
**20193**	9.48 ± 0.53 Aa	1.20 ± 0.01 Cb	2.40 ± 0.06 Bb	<10 UFC/mL
**SP1**	9.51 ± 0.61 Aa	1.82 ± 0.03 Ba	2.35 ± 0.15 Ab	<10 UFC/mL
**T6B10**	9.62 ± 0.39 Aa	1.10 ± 0.06 Cb	2.40 ± 0.04 Ab	<10 UFC/mL
**AM7**	9.40 ± 0.71 Aa	1.00 ± 0.06 Cc	2.30 ± 0.10 Ab	<10 UFC/mL
**P9**	9.21 ± 0.37 Aa	1.82 ± 0.08 Ca	2.50 ± 0.06 Ab	<10 UFC/mL
	** *t30* **			
**Ct**	3.60 ± 0.09 Ac	<10 UFC/mL	4.50 ± 0.21 Aa	<10 UFC/mL
**CA16**	7.90 ± 0.15 Bb	<10 UFC/mL	2.85 ± 0.15 Ab	<10 UFC/mL
**20193**	8.34 ± 0.33 Ba	<10 UFC/mL	2.60 ± 0.17 Ab	<10 UFC/mL
**SP1**	8.12 ± 0.47 Ba	<10 UFC/mL	2.50 ± 0.18 Ab	<10 UFC/mL
**T6B10**	8.33 ± 0.61 Ba	<10 UFC/mL	2.50 ± 0.19 Ab	<10 UFC/mL
**AM7**	8.20 ± 0.52 Ba	<10 UFC/mL	2.38 ± 0.12 Ab	<10 UFC/mL
**P9**	8.11 ± 0.55 Ba	<10 UFC/mL	2.54 ± 0.13 Ab	<10 UFC/mL

The data are the means of three independent experiments (n = 3). Different uppercase letters between different incubation time for the same sample mean significant differences at a *p* value of < 0.005 (*p*, Bonferroni-corrected). Different lowercase letters among different sample within the same time of incubation mean significant differences at a *p* value of <0.00238 (*p*, Bonferroni-corrected).

**Table 3 microorganisms-11-01607-t003:** Acidification kinetics of the YL fermented with *Enterococcus faecium* CA16, *Leuconostoc pseudomesenteroides* DSM20193, *Lacticaseibacillus rhamnosus* SP1, *Lactiplantibacillus plantarum* T6B10, *Levilactobacillus brevis* AM7, and *Weissella cibaria* P9 (CA16, 20193, SP1, T6B10, AM7, P9 are the respective YLs) at 30 °C for 16 h. A: pH difference between inoculation and the stationary phase; Vmax: maximum acidification rate; λ: length of the latency phase expressed in hours.

	A	Vmax	λ
CA16	0.590 ± 0.027 ^d^	0.122 ± 0.008 ^c^	4.363 ± 0.127 ^a^
DSM20193	0.915 ± 0.031 ^b^	0.381 ± 0.016 ^a^	3.360 ± 0.155 ^c^
SP1	0.788 ± 0.054 ^c^	0.113 ± 0.007 ^c^	3.540 ± 0.194 ^c^
T6B10	1.152 ± 0.019 ^a^	0.196 ± 0.008 ^b^	3.690 ± 0.089 ^c^
AM7	0.952 ± 0.048 ^b^	0.182 ± 0.009 ^b^	3.916 ± 0.104 ^b^
P9	0.598 ± 0.012 ^d^	0.103 ± 0.002 ^d^	4.272 ± 0.242 ^a^

The data are the means of three independent experiments. ^a–d^ Values in the same column with different superscript letters differ significantly (*p* < 0.00238, Bonferroni-corrected).

## Data Availability

The data presented in this study are available upon request from the corresponding author.

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
