# Peer review of "Use of Selected Lactic Acid Bacteria and Carob Flour for the Production of a High-Fibre and “Clean Label” Plant-Based Yogurt-like Product"

_microorganisms, 2023, doi:10.3390/microorganisms11061607_

Round 1

Reviewer 1 Report

Overall, the authors present an interesting idea to up-cycle carob flour into a yogurt-like product. The idea is novel and interesting, and the methods are sound. However, the execution in terms of data presentation, statistical analyses, and writing (overall structure, grammar, sentence structuring, discussion) need significant improvement before it can be published.  The potential of carob flour’s role into producing a yogurt like product detail (why is it particularly suited to produce yogurt) should also be elaborated in detail.

Title:

-          The word Yogurt-like should be yogurt-like product, or yogurt-mimic. This should be applied to the entire manuscript.

Abstract:

-          It was difficult to read as the sentences contain too many words and are too long. Sentences should be shortened such that they contain ideally 20-35 words.

-          Line 22: were should be was

-          Keywords should not contain words that are already used in the title

 Introduction:

-          Out of all the different food product possibilities, why was yogurt selected? This was only elaborated in the discussion (lines 397-409). This should be shifted to the introduction. And if any other papers have used carob flour to produce yogurt, the authors should also reference them and how their study differs from existing ones.

Methods:

-          Lines 70-71. Is the carob pulp flour raw or roasted?

-          Lines 72-75. Are the %w/w reported as dry basis?

-          Lines 76-77. Authors should explain the purpose of including rice flour into the formulation.

-          Line 96 Why were the LAB propagated and fermented at 30 °C? The optimum for the 6 LAB listed in lines 78-84 have different optimum temperatures, and vary from 30-45 °C

-          Line 100: log should be Log. cfu should be CFU (and spelled colony forming units at the first mention)

-          Figure 1: 30 gg should be 30 °C. p/v should be w/v.

-          Figure title should include descriptions of abbreviations.

-          Line 127. Latency phase = lag phase? Lag phase is a more common term used to describe microbial growth during fermentation

-          Lines 132-138. The word hours should be replaced with h, to be consistent with the entire manuscript.

-          Line 159. Centrifugation units x g should be consistent with rcf (line 150). Only 1 unit should be used throughout the manuscript.

-          Lines 173-176. Please shorten the sentences. Same for lines 178-183

Results:

-          Table 1. It is incorrect to conduct statistical test across all the samples and t0 and tf. Within t0, 1 way anova should be applied between the 6 LAB and control. Same for samples within tf. Between t0 and tf, a paired t test should be conducted within the same sample. The same should be applied to the other tables in the manuscript.

-          Table 1. Why is lactic and acetic acids units expressed as mmol/L, while for sugars they are expressed as g/L?

-          Table 1. Radical scavenging activity units should be displayed as BHT equivalent or in %.

-          Table 2. Title should italicise the microorganism names.

-          Table 2. What does A mean? Table captions should include abbreviations meaning.

-          Line 265. The authors say that significant decreases were found for sucrose. But the decrease can be considered not practically significant. Besides, the authors should redo their statistical tests according to my 1st point above.

-          Figure 2. Spacing should be added between for16 and 16h.  

-          Line 296. Details should be detail.

-          Line 298. Showed should be show

-          Line 310. Significative should be significant. But again, authors should relook at their statistical method before saying whether their result is significant.

-          In my opinion, Table S1 should be included in the main text, as it was referenced in the main text very often. Key data such as LAB CFU/mL are also in Table S1.

-          Line 364. T6B10 should not be italicised.

-          Tables 1 and 3. Please include standard deviation.

Discussion

-          Lines 400-403. Please shorten sentences.

-          Line 407. "Frequently, plant-based alternatives present high amount of sweeteners". Authors lack citations to back this claim.

-          Line 410. Once should be ones. The word respectively should also be removed.

-          Lines 416-444 should be moved to the introduction.

-          Line 419. The authors mention the need to select a suitable LAB for bioavailability, yet their study does not include any form of bioavailability assessment. The authors should rethink why there is a need to choose a suitable LAB, in the context of their own experiment/ product.

-          Line 445. Who should be which

-          Line 446. Velocity should be replaced with speed

-          Lines 468 to 469 should be reworded.

-          Line 490. The authors mentioned proteolysis as a reason

-          Lines 511-513 should be moved to the introduction.

-          Authors should explain metabolic parameters in detail. For example, why glucose was utilised and not fructose or maltose. Why certain amino acids are utilised, while some are produced. And whether the utilisation/consumption of these metabolites would affect the taste of the product, in relation to their sensory data.

References:

Authors should reduce the number of self citations.

Further proof reading by a native/proficient English speaker is required. Some sentences can be difficult to understand. 

Author Response

Overall, the authors present an interesting idea to up-cycle carob flour into a yogurt-like product. The idea is novel and interesting, and the methods are sound. However, the execution in terms of data presentation, statistical analyses, and writing (overall structure, grammar, sentence structuring, discussion) need significant improvement before it can be published.  The potential of carob flour’s role into producing a yogurt like product detail (why is it particularly suited to produce yogurt) should also be elaborated in detail.

The authors thank the reviewer for the comments. All suggestions have been taken into consideration and the manuscript modified accordingly.

Title:

-          The word Yogurt-like should be yogurt-like product, or yogurt-mimic. This should be applied to the entire manuscript.

The title was modified accordingly as well as the rest of the text.

Abstract:

-          It was difficult to read as the sentences contain too many words and are too long. Sentences should be shortened such that they contain ideally 20-35 words.

-          Line 22: were should be was

-          Keywords should not contain words that are already used in the title

Ok. Sentences were shortened and concepts made clearer. Keywords were updated as recommended.

  Introduction:

-          Out of all the different food product possibilities, why was yogurt selected? This was only elaborated in the discussion (lines 397-409). This should be shifted to the introduction. And if any other papers have used carob flour to produce yogurt, the authors should also reference them and how their study differs from existing ones.

As suggested the explanation of why yogurt-like was chosen was shifted to the introduction and literature updated with recent studies.

Methods:

-          Lines 70-71. Is the carob pulp flour raw or roasted?

The carob pulp used was roasted, it was added in Line 85

-          Lines 72-75. Are the %w/w reported as dry basis?

No, they are expressed as % w/w. Moisture it is also included in the composition.

-          Lines 76-77. Authors should explain the purpose of including rice flour into the formulation.

This aspect was clarified in lines 66-68

-          Line 96 Why were the LAB propagated and fermented at 30 °C? The optimum for the 6 LAB listed in lines 78-84 have different optimum temperatures, and vary from 30-45 °C

Lactiplantibacillus plantarum T6B10, Levilactobacillus brevis AM7, Weissella cibaria P9 and Enterococcus faecium CA16 were isolated from vegetable matrices and their sourdough propagated at 30°C (Rizzello et al. Food Microbiology. 2016, 56, 1-13; Coda et al. Applied and En-vironmental Microbiology. 2008, 74(23), 7391-7398; Montemurro et al. Foods. 2021, 10(2), 462.). Hence, they were routinely propagated at 30°C according to the isolation media and cultivation conditions described in the respective papers, whereas Leuconostoc pseudomesenteroides DSM20193 and Lactobacillus rhamnosus SP1 were propagated at 30°C according to the isolation media and cultivation conditions previously reported. This aspect was clarified in the text (Lines 111-113).

-          Line 100: log should be Log. cfu should be CFU (and spelled colony forming units at the first mention)

Ok. Changes were made throughout the text.

-          Figure 1: 30 gg should be 30 °C. p/v should be w/v.

Figure 1 was updated.

-          Figure title should include descriptions of abbreviations.

Ok. The caption was updated with description abbreviations.

-          Line 127. Latency phase = lag phase? Lag phase is a more common term used to describe microbial growth during fermentation

Yes, usually the lag phase refers to microbial growth, however growth and acidification during fermentation often go at the same pace. Indeed, the sigmoidal shape of the pH profile during fermentation makes it suitable to be modelled using the modified Gompertz or similar equations (Nor-Khaizura et al. 2019. Modelling the effect of fermentation temperature and time on starter culture growth, acidification and firmness in made-in-transit yoghurt. LWT, 106, 113-121) Moreover, the application of the Gompertz model to describe acidification kinetics in yogurt and yogurt-like products is widespread (Soukoulis et al. 2007. Industrial yogurt manufacture: monitoring of fermentation process and improvement of final product quality. Journal of dairy science, 90(6), 2641-2654; Di Cagno et al., 2003. Interactions between sourdough lactic acid bacteria and exogenous enzymes: effects on the microbial kinetics of acidification and dough textural properties. Food Microbiology, 20(1), 67-75.) hence, as reported elsewhere we referred to λ as the latency phase of acidification.  

-          Lines 132-138. The word hours should be replaced with h, to be consistent with the entire manuscript.

Ok. The word was leveled out throughout the text.

-          Line 159. Centrifugation units x g should be consistent with rcf (line 150). Only 1 unit should be used throughout the manuscript.

Ok. The oversight was corrected.

-          Lines 173-176. Please shorten the sentences. Same for lines 178-183

Sentences were shortened and clarified.

Results:

-          Table 1. It is incorrect to conduct statistical test across all the samples and t0 and tf. Within t0, 1 way anova should be applied between the 6 LAB and control. Same for samples within tf. Between t0 and tf, a paired t test should be conducted within the same sample. The same should be applied to the other tables in the manuscript.

The authors thank the reviewer for the comment. Statistical analysis was repeated according to the suggestions and tables and text updated accordingly.

-          Table 1. Why is lactic and acetic acids units expressed as mmol/L, while for sugars they are expressed as g/L?

Usually lactic and acetic acids are expressed as mmol to facilitate the calculation of the fermentation quotient. Regardless, as suggested they were expressed as g/L-mg/L to align all data.

-          Table 1. Radical scavenging activity units should be displayed as BHT equivalent or in %.

As reported in the material and method section, radical scavenging activity was expressed as %. The unit was missing from the tables. They were updated.

-          Table 2. Title should italicise the microorganism names.

Done.

-          Table 2. What does A mean? Table captions should include abbreviations meaning.

A is the pH difference between inoculation and the stationary phase. Table caption was updated.

-          Line 265. The authors say that significant decreases were found for sucrose. But the decrease can be considered not practically significant. Besides, the authors should redo their statistical tests according to my 1st point above.

The reviewer is correct. The statistical was redone as suggested and both text and table updated.

-          Figure 2. Spacing should be added between for16 and 16h. 

Done.

-          Line 296. Details should be detail.

Done.

-          Line 298. Showed should be show

Corrected.

-          Line 310. Significative should be significant. But again, authors should relook at their statistical method before saying whether their result is significant.

Ok. The statical was repeated according to the suggestions.

-          In my opinion, Table S1 should be included in the main text, as it was referenced in the main text very often. Key data such as LAB CFU/mL are also in Table S1.

Table S1 was moved to the main test as suggested.

-          Line 364. T6B10 should not be italicised.

Ok.

-          Tables 1 and 3. Please include standard deviation.

Ok. Standard deviation was included.

Discussion

-          Lines 400-403. Please shorten sentences.

Done

-          Line 407. "Frequently, plant-based alternatives present high amount of sweeteners". Authors lack citations to back this claim.

Reference was added

-          Line 410. Once should be ones. The word respectively should also be removed.

Ok

-          Lines 416-444 should be moved to the introduction.

Done.

-          Line 419. The authors mention the need to select a suitable LAB for bioavailability, yet their study does not include any form of bioavailability assessment. The authors should rethink why there is a need to choose a suitable LAB, in the context of their own experiment/ product.

Indeed, the assessment of minerals or other compounds bioavailability was not performed and should be its own study. Still, fermentation has an influential impact on sensory, nutritional, technological and most of microbial quality of the yogurt-like, hence this aspect was further explained in the discussion (lines 448-457)

-          Line 445. Who should be which

Done.

-          Line 446. Velocity should be replaced with speed

Corrected.

-          Lines 468 to 469 should be reworded.

Ok. The sentence was clarified.

-          Lines 511-513 should be moved to the introduction.

Done.

-          Authors should explain metabolic parameters in detail. For example, why glucose was utilised and not fructose or maltose. Why certain amino acids are utilised, while some are produced. And whether the utilisation/consumption of these metabolites would affect the taste of the product, in relation to their sensory data.

A detail description of the carbohydrate and amino acids catabolism for the strains used in this study and the effect on YL sensory properties was added in lines 498-511 and 556-563.

References:

Authors should reduce the number of self citations.

Self citations were minimized to the essential ones.

Reviewer 2 Report

The authors conducted a study using locust bean meal to produce plant-based fermented beverages. Six LAB strains, isolated from several plant-based raw materials, were used to ferment the rice-bread substrate, and their post-fermentation and shelf-life performance was evaluated by microbiological and biochemical characterisation. A more in-depth characterisation of the nutritional and sensory value of zo was carried out on the strain that showed the best predisposition.

I have a few comments:

1. The word 'yogurt' cannot be used for a beverage based on vegetable raw materials and fermented by these bacterial cultures. The International Dairy Federation and Codex Alimentarius define: yogurt is made from condensed milk and fermented by Streptococcus thermophillus and Lactobacillus delbrueckii ssp. bulgaricus. When we ferment milk by other strains, it is called fermented/probiotic milk. In this case, he proposes to use the name " vegetable fermented beverages" throughout the manuscript. 

2 Lines 102-107 - How much iniculum was added to each trial? What was the size (weight) of each test group during fermentation?

3 Figure1. - The presentation is not as described. This arrangement suggests that the fermentation was with 1 strain. For the CT sample, the 'pathway' should be separated with lactic acid. For what purpose was the CT sample incubated?

4 Lines 132-138 Were the determinations made using the plate method? Under aerobic or anaerobic conditions?

5 Line 124 With which indicator?

6 Line 154. What were the sample dimensions for viscosity assessment?

(7) Lines 211-212. Was the experiment repeated?

8. table 1- According to Fig.1 t0 should be the same for all samples during gelification. You need to make a correction to Fig.1.

9. Table S1- In what units?

10.What do the authors explain the high fructose concentration in sample 20193?

11. Fig.4.- Increase the scale to 10 - now it is 8.

12.The discussion should state which sugars are the carbon source for these strains selected by the Authors?

13.The authors write that the plant-based fermented beverage contains proteins of only 1.58 g/100g, when in cow's milk there is protein of at least 3 g/100g. But is this protein as complete as animal protein? How do the authors assess the amino acid profile of the fermented beverage studied? Can it be a source of essential amino acids?

The authors used 43 references, of which 12 are papers by the authors.

Author Response

The authors conducted a study using locust bean meal to produce plant-based fermented beverages. Six LAB strains, isolated from several plant-based raw materials, were used to ferment the rice-bread substrate, and their post-fermentation and shelf-life performance was evaluated by microbiological and biochemical characterisation. A more in-depth characterisation of the nutritional and sensory value of zo was carried out on the strain that showed the best predisposition.

 The authors thank the reviewer for the comments. All suggestions have been taken into consideration and the manuscript changed accordingly.

I have a few comments:

  1. The word 'yogurt' cannot be used for a beverage based on vegetable raw materials and fermented by these bacterial cultures. The International Dairy Federation and Codex Alimentarius define: yogurt is made from condensed milk and fermented by Streptococcus thermophillus and Lactobacillus delbrueckii ssp. bulgaricus. When we ferment milk by other strains, it is called fermented/probiotic milk. In this case, he proposes to use the name " vegetable fermented beverages" throughout the manuscript.

We completely agree, the word ‘yogurt’ can only be used to define a milk fermented with Streptococcus thermophillus and Lactobacillus delbrueckii ssp. bulgaricus, which is why we did not use the term yogurt but rather “plant-based yogurt-like product” which is a term abundantly accepted in the literature for such products (Levy et al. 2021. Utilization of high-pressure homogenization of potato protein isolate for the production of dairy-free yogurt-like fermented product. Food Hydrocolloids, 113, 106442; Ripari, 2019. Techno-functional role of exopolysaccharides in cereal-based, yogurt-like beverages. Beverages, 5(1), 16; Jiménez‐Martínez et al. 2003. Production of a yogurt‐like product from Lupinus campestris seeds. Journal of the Science of Food and Agriculture, 83(6), 515-522; Yin et al. 2023. Advances in the formation mechanism of set-type plant-based yogurt gel: a review. Critical Reviews in Food Science and Nutrition, 1-20).

2 Lines 102-107 - How much iniculum was added to each trial? What was the size (weight) of each test group during fermentation?

Each trial, which was conducted at least in triplicate, was performed on 100 mL of sample. The inoculum cell density was 7.0 Log CFU/mL, which corresponds to the pellet of roughly 1 mL of LAB culture. This aspect was clarified in the text (Line 116)

3 Figure1. - The presentation is not as described. This arrangement suggests that the fermentation was with 1 strain. For the CT sample, the 'pathway' should be separated with lactic acid. For what purpose was the CT sample incubated?

Fermentation was conducted with 1 strain. Strains were singly inoculated with Enterococcus faecium CA16, Leuconostoc pseudomesenteroides DSM20193, Lacticaseibacillus rhamnosus SP1, Lactiplantibacillus plantarum T6B10, Levilactobacillus brevis AM7, and Weissella cibaria P9 to obtain the following products CA16, 20193, SP1, T6B10, AM7, P9 respectively. The control was incubated in the same conditions as fermented samples to discriminate the effect of the endogenous microbiota, which survived the thermal treatment, from that of the inoculated strains.

4 Lines 132-138 Were the determinations made using the plate method? Under aerobic or anaerobic conditions?

Yes, microbiological analysis were determined using the plate method. Conditions differed for each group determined. Details were added in lines 152-160

5 Line 124 With which indicator?

No indicator was used. pH was measured with a pH meter

6 Line 154. What were the sample dimensions for viscosity assessment?

The viscosity was measured each time on 100 mL of samples. This aspect was added to the text.

(7) Lines 211-212. Was the experiment repeated?

As reported in lines 227-228, three separate sessions were conducted.

  1. table 1- According to Fig.1 t0 should be the same for all samples during gelification. You need to make a correction to Fig.1.

Figure 1 was updated and t0 referred to each sample before fermentation.

  1. Table S1- In what units?

All determinations were expressed as Log CFU/mL. The oversight was corrected.

10.What do the authors explain the high fructose concentration in sample 20193?

The accumulation of fructose when fermenting with EPS producing strains is common since in the presence of sucrose, the strain activates the production of a dextransucrase, which releases fructose (Kothari, D.; Das, D.; Patel, S.; Goyal, A. Dextran and Food Application. In Polysaccharides: Bioactivity and Biotechnology; Springer: Cham, Switzerland, 2015; pp. 735–752). This aspect was clarified in the discussion (Line 506-510).

  1. Fig.4.- Increase the scale to 10 - now it is 8.

Ok. Figure 4 scale is from 0 to 10.

12.The discussion should state which sugars are the carbon source for these strains selected by the Authors?

A detail description of the preferred carbon sources for the strains used in this study was added in lines 498-510

13.The authors write that the plant-based fermented beverage contains proteins of only 1.58 g/100g, when in cow's milk there is protein of at least 3 g/100g. But is this protein as complete as animal protein? How do the authors assess the amino acid profile of the fermented beverage studied? Can it be a source of essential amino acids?

Free amino acids were determined for all fermented samples and all essential amino acids were present in the selected beverage. However, as the reviewer pointed out and as specified in the discussion section (Lines 440-444), the formulation proposed in this study lacks the amount of protein that a conventional yogurt contains. Which is why further studies should focus on the optimization of the formulation to have a better nutritional profile.

The authors used 43 references, of which 12 are papers by the authors.

Self citations were minimized to the essential ones.

Round 2

Reviewer 2 Report

The authors have revised the manuscript. I have no further comments.